# Tunable photo-responsive elastic metamaterials

Antonio S. Gliozzi [1✉], Marco Miniaci [2,3], Annalisa Chiappone[1], Andrea Bergamini [3], Benjamin Morin [3] & Emiliano Descrovi[1,4]

The metamaterial paradigm has allowed an unprecedented space-time control of various physical fields, including elastic and acoustic waves. Despite the wide variety of metamaterial configurations proposed so far, most of the existing solutions display a frequency response that cannot be tuned, once the structures are fabricated. Few exceptions include systems controlled by electric or magnetic fields, temperature, radio waves and mechanical stimuli, which may often be unpractical for real-world implementations. To overcome this limitation, we introduce here a polymeric 3D-printed elastic metamaterial whose transmission spectrum can be deterministically tuned by a light field. We demonstrate the reversible doubling of the width of an existing frequency band gap upon selective laser illumination. This feature is exploited to provide an elastic-switch functionality with a one-minute lag time, over one hundred cycles. In perspective, light-responsive components can bring substantial improvements to active devices for elastic wave control, such as beam-splitters, switches and filters.

[1] Department of Applied Science and Technology, Politecnico di Torino, Corso Duca degli Abruzzi 24, 10129 Torino, Italy. [2] CNRS, Univ. Lille, Ecole Centrale, ISEN, Univ. Valenciennes, IEMN - UMR 8520, 59046 Lille cedex, France. [3] Empa, Laboratory of Acoustics and Noise Control, Überlandstrasse 129, 8600 Dübendorf, Switzerland. [4] Department of Electronic Systems, Norwegian University of Science and Technology, O.S. Bragstads plass 2b, 7034 Trondheim, Norway. ✉email: antonio.gliozzi@polito.it

Elastic metamaterials achieve their superior unusual properties thanks to a collective interplay of the local properties of individual artificial structural elements. In particular, the propagation of elastic waves through periodic structures is governed by the elastomorphic and material parameters of the unit cell[1–4]. By properly designing the unit cell size, shape and spatial arrangement, specific elastic functionalities can be achieved, such as frequency band gaps[5,6], focusing beyond the diffraction limit[7,8], defect-immune and scattering-free wave propagation[9–11], as well as cloaking[12,13]. Despite the high potential demonstrated so far, most of the proposed elastic metamaterials are lacking in dynamic tunability, meaning that they cannot adapt to varying environmental conditions or external stimuli. To address this issue, several approaches have been proposed so far, based on piezoelectric[14–17], temperature-[18–21] and magnetic field-sensitive materials[22–25]. External mechanical loads that directly deform the unit cell have also been used[26,27], in some cases leading to auxetic or to bi-stable metamaterials[28]. Similarly, dynamically reconfigurable properties[29], loss-compensating materials[30] and parity-time-symmetric configurations[31] have recently shown the possibility to achieve an anisotropic transmission and to break mechanical reciprocity with an active control of the system[17]. However, while these approaches are extremely attractive for selectively tuning of the material effective density and stiffness under controlled conditions[32], they typically involve a significant increase of complexity in practical implementations (e.g., the need for an additional electrical wiring, mechanical stability and physical contact requirements). Very recently, Walker and coworkers have proposed active elastic metamaterials made of nanocomposites susceptible of infrared (IR) or radio-frequency (RF) activation[33]. For example, in ref. [34], the authors show the reversible opening of a transparency window obtained after RF irradiation lasting several tens of minutes. The use of electromagnetic (EM) radiation is very attractive as it can be provided in a contactless fashion, with arbitrary intensity, frequency and polarization state patterns. This is particularly straightforward to accomplish when visible light is preferentially employed, thanks to the large availability of sources and optoelectronic devices for controlling radiation wavelength, polarization, spatial, and angular distributions.

In parallel, recently published works have shown that photocurable polymers containing azobenzene units can be succesfully exploited in 3D printing processes to fabricate devices with light-triggered functionalities. Azopolymers are well-known molecular photo-switches whose photo-isomerization process constitutes the underlying mechanism for many photo-induced effects, including photo-fluidization, mass-flow and surface relief formation[35–37], and also mechanical photo-actuation[38–43]. As an exemplary host polymeric matrix, a diacrylate monomer Bisphenol A Ethoxylate Diacrylate (BEDA) has been used in combination with proper photoinitiators, such that a radiation-assisted crosslink process can be carried out, for instance, in a Digital Light Processing (DLP) 3D-printing or two-photon lithography system. Once crosslinked, the BEDA matrix constitutes a passive mesh embedding azopolymer units that are eventually subjected to a photo-isomerization triggered by visible light. The photo-isomerization of the azo-units promotes an efficient energy transfer to the passive polymeric matrix[44], thus inducing a large Young's modulus change that significantly surpasses the commonly observed thermal-softening of glassy polymers[45]. In previous papers, amorphous azo-doped BEDA microstructures fabricated by DLP or two-photon lithography have been demonstrated to exhibit reversible light-induced volume and refractive index changes[46] and either photo-softening or photo-hardening at temperatures below or above the polymer Tg (i.e., the glass transition temperature, at which polymers gradually change from a hard and relatively brittle "glassy" state into a

rubbery state), respectively[47]. These effects can be significant: in a BEDA matrix filled with a Disperse Red 1 Methacrylate (DR1M) azopolymer, a 75% relative decrease of the Young's modulus has been measured upon green light illumination. When the illumination is switched off, the original value of the Young's modulus is fully recovered. Contrary to azo-doped liquid crystalline elastomers, the Young's modulus variation in such disordered materials is isotropic and substantially localized to the irradiated area, with a limited long-range effect, also due to the low thermal conductivity of the polymeric matrix.

Here, we introduce a light-responsive elastic metamaterial consisting of a linear array of squared pillars arranged on a slab waveguide (Fig. 1a). The metamaterial is fabricated via UV-polymerization of Bisphenol A Ethoxylate Diacrylate (Mn 512, BEDA, Fig. 1b) and a BAPO photoinitiator in a DLP system. Methyl red (MR) is employed as an azo-dopant, serving both as an optical absorber to ensure a good spatial resolution during the printing process and as an active light-responsive component (Fig. 1c). In the following, we show that the azo-unit photoisomerization can be exploited to trigger a controlled decrease of the Young's modulus and a corresponding modification of the finite waveguide elastic response. In particular, we demonstrate that the vibration eigenmodes of individual resonant pillars are affected by a local illumination in such a way that the overall metamaterial band gap can be significantly widened. These effects are shown to occur within a minute time range and are fully reversible. Finally, we provide an exemplary application of the presented light-responsive metamaterial as a single-frequency active filter, whose repeatibility has been checked for a hundred cycles.

## Results

**Light-Responsive MetaMaterial: geometry and elastic features.** The proposed Light-Responsive MetaMaterial (LRMM) waveguide is $h = 500\,\mu m$ thick and covers a total length of $L = 60\,mm$. Along the waveguide, a periodic linear array of eight square pillars is arranged over a length of 40 mm (Fig. 1d). The crystal unit cell includes a single pillar placed on top of the slab, as shown in Fig. 1e, and is characterized by a lattice parameter of $a = 5\,mm$. The pillars have a height $H = a$ and lateral size $b = 0.75a$. Elastic waves are excited into the waveguide by means of a piezoelectric transducer glued to one side of the slab (as shown in Fig. 1f). During the wave propagation phenomenon, the out-of plane velocity is measured at different points by means of a Scanning Laser Doppler Vibrometer (SLDV) operating at 633 nm. An additional 405 nm wavelength laser is used to trigger light-induced variations of the Young's modulus of the material, with an overall available power of up to 250 mW, focused onto an area of about 30 mm². This allows the selective illumination of individual pillars. In order to study the spectral response of the structure, a linear frequency sweep of $t = 500\,ms$ is fed to the piezo-actuator, so that waves within a 25–125 kHz frequency range are excited. As the elastic waves injected by the piezo source travel along the structure, pillars are progressively put into oscillation, absorbing energy according to their individual eigenmodes. The evolution of the transmission spectrum along the propagation direction, during the frequency-swept excitation, can be clearly observed in Fig. 1g, reporting the Fourier-transformed vibration signals of each pillar, measured at the center of its top surface. The cumulative effect of the 8-pillar array in absorbing the elastic wave energy results in the opening of a frequency band gap (BG) from 70 to 85 kHz, as measured at a point downstream of the pillar array (see the spectrum labeled as "out" in Fig. 1g). The frequency- and momentum-resolved repartition of the injected elastic energy among the allowed modes of the LRMM can be evaluated by

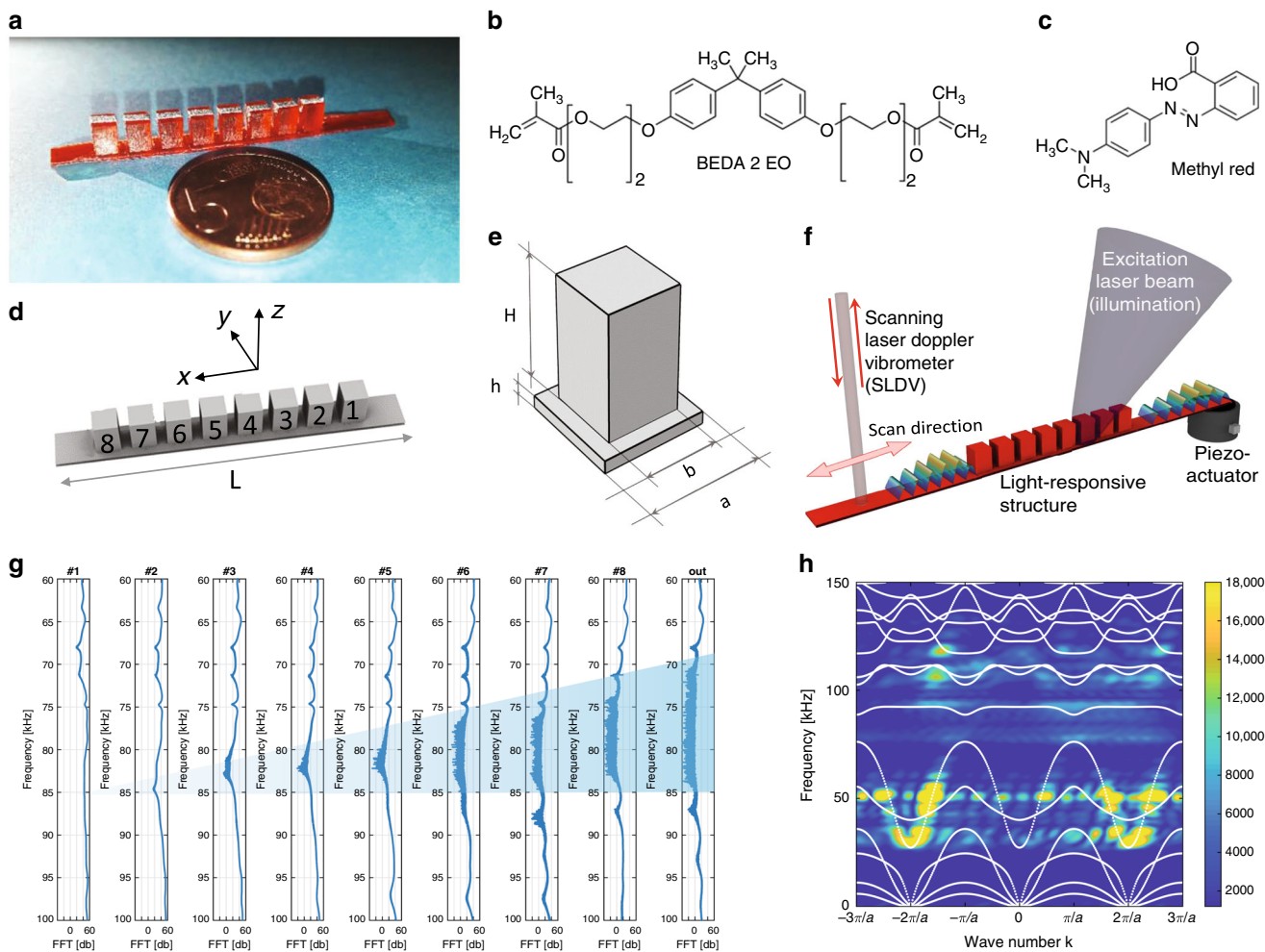

**Fig. 1 LRMM concept and spectral response without illumination. a** Picture of the LRMM patterned with periodic square pillars. **b** Molecular structures of the azopolymer and **c** the UV-curable resin used for fabrication. **d** Isometric view in the x-direction of the numerical design of the waveguide. **e** Schematic representation of the unit cell of the periodic pattern: $a = 5$ mm, $H = a$, $h = a/10$ and $b = 0.75a$. **f** Sketch of the experimental configuration in which a piezo-actuator excites elastic waves in the slab, a 405 nm laser beam is used to induce a spatially selective photo-softening effect depending on the illumination area and a 633 nm laser vibrometer scans the whole structure while detecting point-by-point the out-of plane oscillation amplitude (SLDV). **g** Spectral response measured along the pillar array, when elastic waves within a broadband frequency range are excited at one edge of the specimen. A clear progressive drop of the oscillation amplitude on top of each individual pillar is observed at frequencies within the band gap (blue-shaded region), as the scanning laser of the vibrometer moves toward the edge opposite to the excitation source. **h** Measured dispersion curves of the photo-responsive elastic waveguide at without illumination, superposed to the calculated band structure (white lines). The design of the unit cells allows the nucleation of a frequency band gap centered at about 80 kHz.

performing a temporal and spatial Fourier Transform (2D-FT) of signals collected along the longitudinal x-direction of the pillar array. To this end, signals are collected along a line, with a spatial resolution of 0.1 mm during the transient propagation of elastic waves injected according to a Hanning-modulated sine function (3 cycles) centered at $f_0 = 75$ kHz. Results are presented in Fig. 1h as a color map indicating the frequency and momentum coordinates in which the elastic energy is mainly distributed over the first three Brillouin zones. Numerically calculated dispersion curves are superimposed to the experimental measurements as white lines, revealing the elastic wave propagation occurs according to the pass bands supported by the patterned waveguide, while a negligible elastic energy is propagated within the BG frequency range.

**Tuning and recovery of the LRMM spectral response.** Each pillar contributes to the overall structure response as an individual oscillator, whose multiple eigenmodes are determined by

pillar size, geometry and material properties, namely density and elastic modulus, chosen identical for each pillar in our case.

The photo–softening effect mentioned above is first observed on a single pillar by measuring the corresponding illumination-dependent frequency response. More specifically, the out-of-plane velocity of a propagating wave generated through an excitation with a flat spectrum in the 75–95 kHz frequency range at the input slab edge is collected at the center of the upper surface of pillar #1, during laser exposure. As shown in Fig. 2a, the pillar spectral response (i.e., the FFT amplitude of the signals reported on the z-axis of the figure) is first measured without illumination (red curve) and then at increasing laser power (10 mW steps). A frequency down-shift of the principal resonance peak located at 81 kHz is clearly observed, accompanied by a progressive attenuation of the oscillation amplitudes at higher frequencies (around 90 kHz), for increasing illumination intensities. The down-shift and amplitude attenuation of the resonances are expected after the polymer photosoftening.

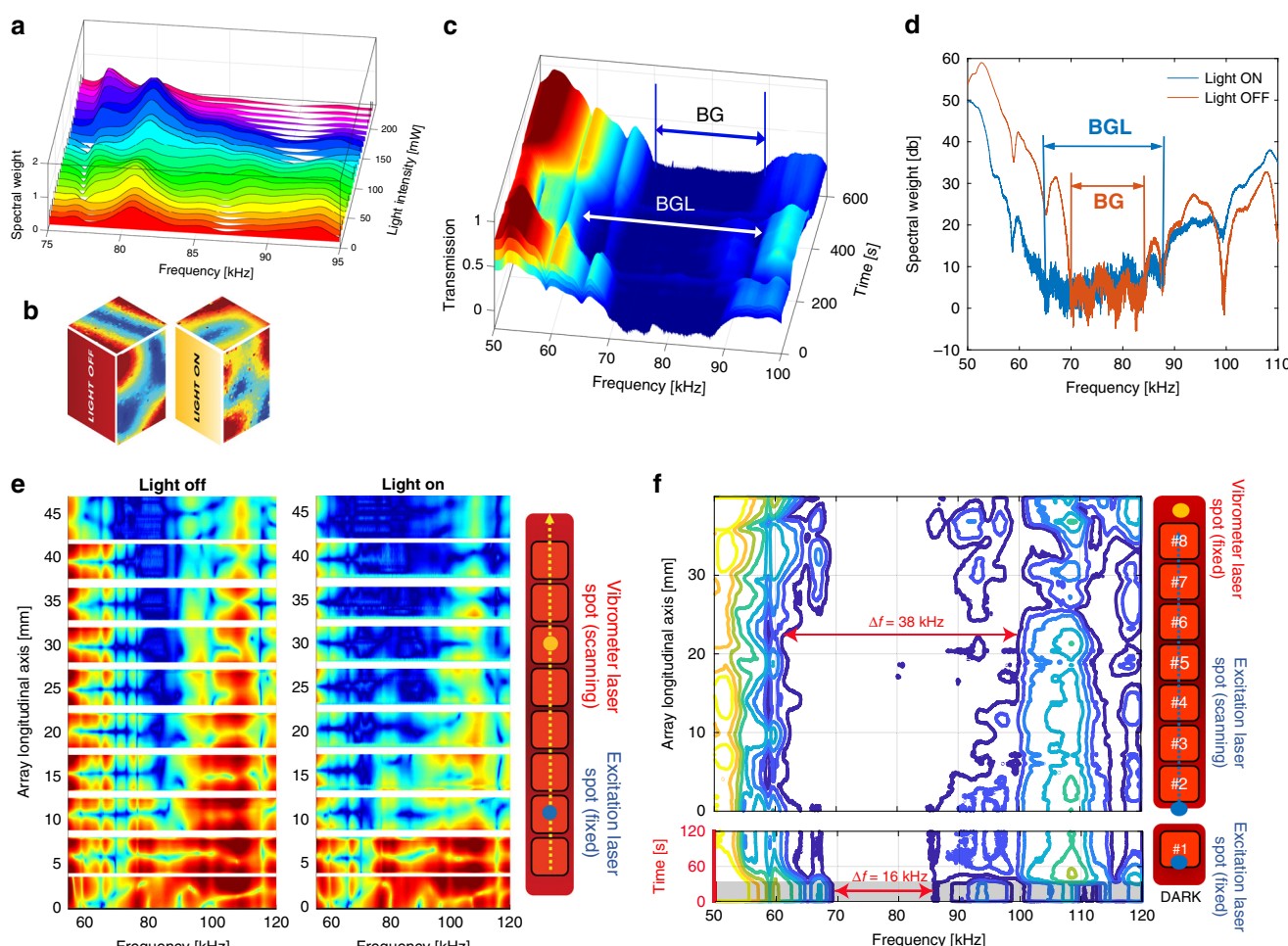

**Fig. 2 Photo-induced tunability of the LRMM frequency response. a** Spectral response of pillar #1 in the range 75–95 kHz under illumination at increasing intensity. Measurements are performed at the center of the top surface of the pillar. **b** 3D representation of the displacement distribution over pillar #1 surface (top and lateral) produced by a monochromatic elastic wave excitation at 98 kHz, without and with illumination at 200 mW laser power. **c** Temporal evolution of the LRMM transmission spectrum under laser illumination. At $t = 100$ s, the laser beam is switched ON and selectively directed on pillar #1 only. Then, at $t = 360$ s the laser is switched OFF. The normalized amplitude of the out-of-plane elastic wave oscillations is collected as a function of time, at the waveguide edge opposite to the piezo source. A light-induced band gap (BGL) can be observed as a result of a widening and a down-shift of the lower edge of the original band gap (BG). The change of the spectral response can be appreciated almost instantaneously, but a time interval of about 100 s is required to reach a steady state. **d** Transmission spectrum of the LRMM upon a broadband excitation without illumination (red curve) and during the illumination of the first pillar, once a steady state is reached (blue curve). **e** Spatially resolved spectral response of the LRMM without illumination (left panel) and under illumination of pillar #2 only (right panel) obtained by SLDV along the longitudinal x-axis of the sample. **f** Contour plot of the LRMM transmission spectrum measured at an output point at the end of the waveguide during a scan of the excitation laser from pillar #1 to pillar #8. During the first 30 s, spectra are collected without illumination, for reference purposes, then pillar #1 is illuminated while keeping constant the position of the excitation laser for $t = 100$ s. Next, the excitation laser is slowly moved so that each pillar is sequentially illuminated. The frequency band of the BG broadens from 16 kHz without illumination to 38 kHz corresponding to the illumination on pillar #6. Colors from blue to red indicate an increasing amplitude of the measured out-of-plane displacement.

However, a single pillar can feature less straightforward spectral modifications, depending on the illumination conditions. Indeed, the light-induced changes of the elastic modulus are inhomogeneously distributed within the pillar volume (because of the material absorption, laser beam focusing, illumination angle, etc.). This affects each local eigenmode differently, in terms of eigenfrequency, Q-factor and spatial distribution, possibly leading to complicated inter-mode energy coupling. Figure 2b provides experimental evidence of such an effect by showing the distribution of the maximum displacement amplitude over a pillar surface, before and during laser irradiation. A continuous monochromatic elastic wave at 98 kHz is used here to excite pillar #1, while the SLDV scans the top and lateral pillar surface. As pillar #1 is illuminated by the excitation laser (in this case the

laser power is set at 200 mW), the spatial distribution of the maximum displacement amplitude is strongly modified in response to a new configuration for the energy distribution among the available eigenmodes.

Despite the illumination of pillar #1 is local, the overall frequency response of the LRMM is significantly altered. In Fig. 2c, we illustrate the temporal evolution of the transmission spectrum of the LRMM as the laser excitation on pillar #1 is switched on and then off. The piezo-source is continuously fed with a sinusoidal signal, with a constant amplitude $A_{inp,1} = 3$ V$_{pp}$ and a sweeping frequency ranging from 45 to 125 kHz, over repeated cycles of $\Delta t = 0.5$ s. The SLDV measures the out-of plane velocity in a fixed point on the opposite edge of the slab, downstream of the pillar array. Measurements are performed at

intervals of 1 s. During the first 100 s, no laser illumination is provided to the sample. Then, pillar #1 is selectively illuminated with a 150 mW laser beam. After few tens of seconds, the low-frequency edge of the BG is observed to down-shift by 7 kHz, with a corresponding widening of the BG up to 10 kHz. This effect is illustrated in Fig. 2d, where the transmission spectrum before and after a 100 s illumination is shown. After $\Delta t_m = 260$ s the laser is switched off, and the LRMM recovers its initial spectrum. A stationary condition is typically reached after about 100 s of illumination at 150 mW and then maintained if the illumination power is kept constant. In comparison to other tuning approaches recently reported (e.g., 50 min in ref. [34]) the time delay for the BG variation appears minimal (an ~30 fold reduction for a full cycle).

**Local illumination and heating effects on the LRMM spectrum**. The effect of a local laser illumination on the band gap tuning can be better understood by looking at the evolution of the wavefield spectrum during the wave propagation through the pillar array. To do this, measurements are performed by spatially scanning the structure along the longitudinal x-axis, with and without laser illumination. In Fig. 2e, normalized wavefield spectra at different locations along the x-axis are presented as log-scale colormaps. The spatial resolution of the SLDV is $\Delta x = 0.1$ mm and data collection is performed at a sampling rate of 1 Sa/s. In this case, only pillar #2 is illuminated (the laser power is set to 200 mW), thus leaving the spectrum measured on pillar #1 to serve as a reference. Differences in the wavefield are seen to be significant in transmission only, downstream of pillar #2, while almost no modifications due the reflected wave are observed on the spectrum measured upstream of pillar #2. More importantly, the band gap broadening occurs mainly in a frequency range in which single pillar resonances down-shift under laser illumination (see Fig. 2a). As discussed below, these observations suggest that the underlying mechanism for the light-induced tuning of the band gap is strongly associated to a modification of the single pillar local resonances, which seems to prevail on other possible collective Bragg-type effects. With a 200 mW laser power, the isomerization-driven photothermal effect is such that the irradiated pillar reaches a stationary temperature of about $T = 45°$, as illustrated in the Supplementary Fig. 1. At this temperature, which remains lower than the glass-transition temperature Tg $\simeq$ 60°, the Young's modulus of a BEDA-MR thin slab measured at 80 kHz decreases by roughly 28%, namely from 1.75 GPa at room temperature to 1.25 GPa (see Supplementary Fig. 2). No significant changes in temperature are observed during laser irradiation of a BEDA slab without azo-dopants.

The laser illumination can be directed on specific locations of the structure, thus providing an additional degree of freedom in controlling the LRMM overall spectral response. In order to illustrate this aspect, we measured the LRMM transmission spectrum at a fixed location of the SLDV at the output end of the waveguide while sequentially illuminating all eight pillars along the array. The piezo-source injects a frequency-modulated signal, with a uniform spectral content in the range 45–125 kHz and a constant amplitude of 3 $V_{pp}$. Results are presented in Fig. 2f as a two-dimensional contour plot, where the vertical axis indicates the laser excitation position along the longitudinal x-axis of the waveguide. Initially, spectra are collected without illumination at a rate of 0.5 Hz, displaying the previously shown 16 kHz–wide band gap. After 30 s, the laser is switched on, at a starting position corresponding to pillar #1. The irradiation is kept constant for 100 s and the spectral evolution is recorded, until the steady state of the light-induced change of the elastic modulus is reached. Then, the exciting laser is displaced very slowly along the

waveguide at spatial steps of $\Delta x = 0.05$ mm and the output spectrum is recorded at a rate of 0.5 Hz for each position of the excitation laser beam. A sufficiently slow speed of the illuminating laser is chosen to avoid cumulative effects caused by the contribution of pillars that have been previously illuminated but still have not relaxed back completely, after the laser spot has moved forward. The evolution of the transmission spectrum can be readily appreciated in terms of a progressive broadening of the band gap and changes in the relative spectral weights. Both band edges move further apart, leading to a stop band reaching a 38 kHz width when the illumination is provided on pillar #6 (Fig. 2f).

In order to better investigate the mechanism exploited in the band gap modification, we show by means of thermal imaging of the samples (see Supplementary Fig. 2) that laser illumination induces very localized heating of the azo-doped structure. The temperature enhancement observed typically corresponds to a Young's modulus reduction of 28%, as can be verified by the Dynamic Mechanic Analysis (DMA) measurements reported in the Supplementary Note 2. The local softening discussed above has an effect that is considerably different from the one obtained by uniformly softening the whole material (see Supplementary Fig. 3c), which is what is commonly done in the literature when environmental conditions (e.g., temperature) are changed or the structure is irradiated with non focused fields[18,33,34]. In the case of uniform heating of the structure, we expect that each resonant unit undergoes the same spectral transmission change, resulting in a more or less rigid red-shift of the BG. Instead, performing a selective softening of the resonating units, each pillar acts as an individual mechanical resonator with its own vibrational eigenmode spectrum, so that we speculate that the observed global transmitted spectrum is due to the overlap of the original lattice spectrum with the one of the softened resonator(s).

This is confirmed by three-dimensional finite element (FE) model calculations of the LRMM transmission spectrum when a progressive number of pillars (from #2 to #8) undergoes a Young's modulus reduction (see Supplementary Note 4 for details). In agreement with experiments, an overall widening of the band gap is predicted when only few pillars are softened (e.g., pillars #2–#3), while frequencies outside the band gap are not significantly affected. On the contrary, when the softening is applied to the large majority of the pillars (e.g., pillars #2–#8), their corresponding eigenmode spectra again become overlapped, leading to the overall redshift of the band gap. These results point towards a mechanism mainly related to local resonance, although Bragg-type scattering effects cannot be ruled out[48,49].

**Light-triggered transmission filter**. The light-induced tunability of the LRMM response suggests several applications. We propose here the example of an active acoustic filter with tunable attenuation. According to the results shown above, the transmission of an elastic wave at a specific frequency can be easily controlled by means of an appropriate laser illumination. We consider a configuration in which a sinusoidal wave at 95 kHz is fed into the sample as an input signal at one edge and the transmitted amplitude is monitored as output at the opposite edge. In Fig. 3a, the time trace of the maximum transmitted amplitude is presented during 6 on-off illumination cycles on pillar #2, with a constant power (150 mW). As the input wave falls in proximity of the frequency band gap edge, it undergoes attenuation as a result of the laser-induced band gap widening described above. When the excitation laser is switched on, a significant drop of the transmitted wave amplitude is observed, followed by a full recovery of the signal when the light is switched off. The relative amplitude here is reduced by more than 50%

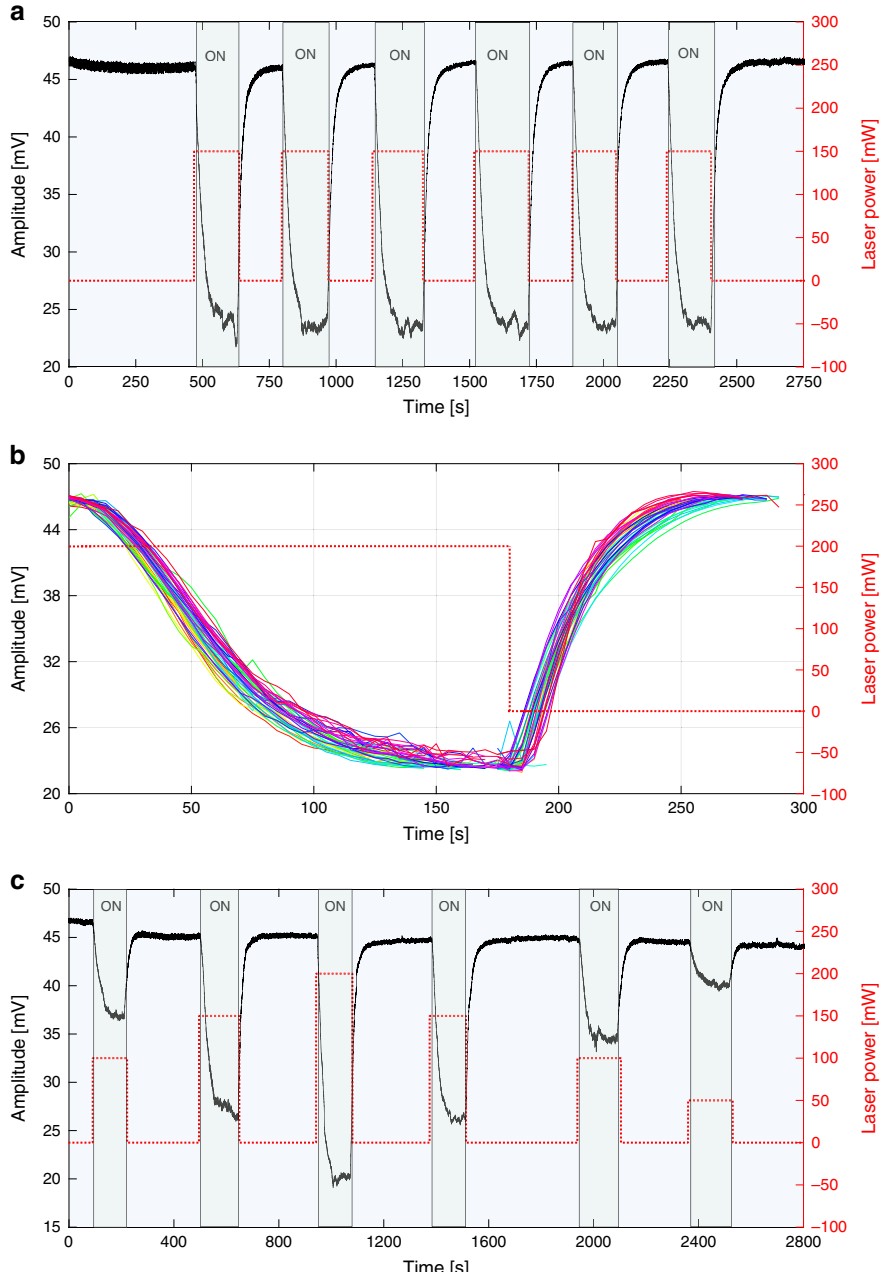

**Fig. 3 Photo-switchable elastic filter. a** Amplitude of the elastic wave transmitted through the LRMM during 6 on-off illumination cycles with a 150 mW laser power. **b** Temporal trace of the on-off switching over 100 cycles. The effect is reversible, with the recovery dyamics being described by a mono-exponential function with a $\tau \simeq 67$ s time constant. Different colors represent different repetitions of the same on-off cycle (only 50 cycles are reported for clarity). **c** Light-induced modulation of the transmitted acoustic wave amplitude at several illumination power levels. All measurements are performed at a frequency of 95 kHz, at the upper frequency edge of the light-induced band gap (BGL).

(6 dB). The process reproducibility has been tested up to 100 on-off cycles, as shown by the time-resolved measurements in Fig. 3b (the different colors highlight the different on-off cycles) thus demonstrating a low degradation of the overall material light-responsivity at these irradiation doses. The new elastic state reached after illumination displays stable conditions in time and no further changes are expected, due to e.g. energy accumulation, until the light is switched-off. In Fig. 3c, the time-resolved transmitted amplitude during several on-off illumination cycles for varying power levels is presented. In addition to the control on the frequency domain, the transmitted wave can also be amplitude-tuned by varying the excitation laser intensity. It is interesting to note that the recovery dynamics, from the

Low-transmission level $A_L$ (light on) to the High-transmission level $A_H = A_L + \Delta A$ (light off) in all time traces in Fig. 3a, b, can be adequately described by a time-varying amplitude function $A(t) = A_L + \Delta A(1 - \exp(-t/\tau))$, where $\tau \simeq 67$ s is a time constant independent from $A_L$ or, in other words, the light excitation intensity. In the illumination conditions considered in Fig. 3c, the relative amplitude attenuation $\Delta A/A_H$ scales proportionally with the laser power.

## Discussion

In conclusion, we have presented a light-responsive metamaterial realized through a 3D printed patterned waveguide. Experimental evidence illustrates the potential use of user-defined light irradiation

patterns to control the elastic response of these structures, in particular, to tune the width of a frequency band gap. This new approach presents several advantages over other known techniques, such as contactless control, full switching reversibility and quick response (about a minute to reach a steady state). The proposed structure constitutes only a specific example of a new class of active elastic metamaterials that can be accessible by applying 3D printing techniques to photocurable polymers containing light-active units. Depending on the absorption spectrum of the latter, the light-responsivity can be preferentially improved at other optical wavelengths (e.g., at 532 nm, if DR1M azo-dopant is employed). We believe these results will open new exciting opportunities in elasticity and acoustics, with the possibility of obtaining novel effects and unprecedented degrees of control thanks to the application of light fields with complex spatio-temporal distributions to existing active metamaterial platforms.

## Methods

**Simulations.** Dispersion diagrams and mode shapes presented in Fig. 1h are computed using Bloch-Floquet theory in full 3D FEM simulations carried out via the Finite Element solver COMSOL Multiphysics. Full 3D models are implemented to capture all possible wave modes. A linear elastic constitutive law is adopted and the following mechanical parameters used: density $\rho = 1150$ kg/m$^3$, Young's modulus $E = 1.75$ GPa, and Poisson's ratio $\nu = 0.34$. Domains are meshed by means of 8-node hexaedral elements of maximum size $L_{FE} = 0.5$ mm, which is found to provide accurate eigensolutions up to the frequency of interest[50]. The band structure is obtained assuming periodic conditions along the lattice vectors $a_1$. The resulting eigenvalue problem $(\mathbf{K} - \omega^2\mathbf{M})\mathbf{u} = \mathbf{0}$ is solved by varying the non-dimensional wavevector $\mathbf{k}$ along the $[\Gamma - X]$ boundary of the first irreducible Brillouin zone, with $\Gamma \equiv (0, 0)$, $X \equiv (0, \pi/a)$.

**Fabrication.** Bisphenol A ethoxylate (2 EO/phenol) diacrylate (Mw 572, BEDA), 2-(4 dimethylaminophenylazo) benzoic acid (methyl red, MR) and Bis(2,4,6-tri-methylbenzoyl)-phenylphosphineoxide (BAPO) are purchased from Sigma-Aldrich. The dye (MR) is directly dispersed in the liquid monomer (0.1 per hundred resins, phr) and sonicated for 30 min to obtain an homogeneous dispersion. Then BAPO, previously dissolved in acetone (concentration in acetone 80 mg ml$^{-1}$) is added. The formulation is printed using a commercial DLP printer HD 2.0 (Robot Factory), equipped with a projector (resolution $1920 \times 1080$ pixels) with a light intensity of 10 mW/cm$^2$ (measured with a Hamamatsu Power meter). The layer thickness is set at 20 μm and the exposure time is 3 s/layer for the base structure and 2 s/layer for the pillars. The printed samples undergo a post curing process (2 min) performed with a medium pressure mercury lamp equipped with a rotating platform (intensity 10 mW/cm$^2$, Robot Factory).

**Experimental measurements.** The specimen, consisting of 8 unit cells, is fabricated through a two step process. The specimen is made of a photocurable resin containing an azopolymer additive, with the following measured properties: density $\rho = 1150$ kg/m$^3$, Young's modulus $E = 1.78$ GPa, and Poisson ratio $\nu = 0.34$. The geometrical parameters are the following: $a = 5$ mm, $H = a$, $L = 0.75 \cdot a$. Elastic waves are excited through a piezoelectric IMG transducer, bonded to the bottom surface of the slab at the as in Fig. 1f. The experimental wavefields are measured using a scanning laser Doppler vibrometer (SLDV) that measures the out-of-plane velocity of points belonging to a predefined grid over the structure. The spatial resolution of the measurements can be varied according to the spatial resolution of the linear stage (here the minimum used is 0.05 mm).

## Data availability

The data that support the plots within this paper and other findings of this study are available from A.S.G. and E.D. upon reasonable request.

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

## Acknowledgements
This project has received funding from the European Union's Horizon 2020 FET Open ("Boheme") under grant agreement No. 863179. M.M. has received funding from the European Union's Horizon 2020 research and innovation programme under the Marie Skłodowska-Curie grant agreement N. 754364. Part of the present work has been performed at POLITO BIOMed LAB. We are grateful to Prof. M. Scalerandi and Dr. F. Bosia for critical reading of the manuscript and discussion.

## Author contributions
A.S.G. and E.D. conceived the idea and coordinated the work; M.M. defined the structure design; M.M. and A.B. performed the numerical calculations; A.C. developed the polymeric compound and fabricated the structure; A.S.G. built the experimental set-up and performed the measurements; A.S.G and E.D. performed data analysis; B.M. and M.M. performed the DMA to characterize the temperature-frequency-dependent mechanical properties of the polymer. All the authors contributed to the manuscript writing.

## Competing interests
The authors declare no competing interests.
