## [Peer Review File · Nature Communications]

Reviewers' comments:

Reviewer #1 (Remarks to the Author):

This manuscript presents experimental results on thermal modification of acoustic band structure of a linear periodic chain of elastic scatterers. The scatterers are fabricated from a thermo-sensitive polymer which becomes softer under laser heating. The authors demonstrated relatively fast and reversible response of the chain by measuring the transmissivity of the chain subjected to a wide spectrum of elastic vibrations. The experiments are carried out with high precision allowing detection of the amplitude of vibrations within a very small area of each scatterer. The obtained results can find applications in design of remote controlled acoustic filters. I consider the presented results as a high-quality research, which, however, does not fit the scope of Nature Communication. The authors do not propose a new mechanism of control of the band structure; they rather propose a useful modification of the already known methods.

I do not agree with the authors' interpretation of the obtained results. The authors claim that heating of a single scatterer leads to extension of the band gap. This is not true. If elastic properties of a single element in a periodic chain are different from the others, this element is considered as a point defect in a periodic system. It is known that presence of a point defect leads to a discrete level within the band gap (so-called impurity level). If the level appears close to one of the band edges and the level width (due to dissipation) is sufficiently broad, it can be mistakenly interpreted as broadening of the band gap. There is one more reason for the observed extension of the gap. The band edges are not sharp because the chain contains only 8 periods. If the last element in the chain is removed it becomes shorter that leads to further degradation of the band gap. If the element is not removed completely but replaced by a defect, a Tamm state may appear in the spectrum. Since the chain is short the level associated with this state may be sufficiently broad that, again, can be interpreted as gap broadening. Additional study is necessary in order to understand what is the real physical nature of the observed lowering of acoustic transmissivity.

In conclusion, I do not recommend this paper for publication in Nature Communication since the presented results do not contain a new physical effect and the proposed explanation is confusing.

Reviewer #2 (Remarks to the Author):

Authors presents the first steps of stimuli responsive metamaterials. The paper is well written and consistant. From the physical point of view i do not see any mistake. I cannot judge on the chimsitry but the obtained results confirm the efficiency of the procedure.

In overall this study is perfectly in time with the current state of art to reach time modulated active metamaterial.

I do recommend the paper for publication.

Minor: they should discuss more the maximum modulation time of such structures and fix the typos in references.

Reviewer #3 (Remarks to the Author):

Title: Tunable photo-responsive acoustic metamaterials
by A. S. Gliozzi et al

The authors describe a one-dimensional acoustic metastructure based on a 3D printed light-responsive patterned waveguide. Evidence is given on how to tune the width of the frequency

band gap. Undoubtedly the paper is interesting and well written, but some figures are unclear and needs to be revised.

In Conclusion the manuscript can be considered for publication on Nature Communications but only after some mandatory major revisions according to the following critical points:

Figure 2a: the authors should define what is the "spectral weight" on the z axis of the figure.

Figure 2f: it is unclear. On the vertical axis two different quantities are mixed: Time [ms] and Array longitudinal axis [mm]. This may confuse the general reader.

Figure 3b: Dotted red line is interrupted. Moreover the difference among the colour curves are not reported in the caption. The authors should clarify this point.

Line 74: the authors should explain in the text the meaning of the quantity T_g

Reviewer #1

This manuscript presents experimental results on thermal modification of acoustic band structure of a linear periodic chain of elastic scatterers. The scatterers are fabricated from a thermo-sensitive polymer which becomes softer under laser heating. The authors demonstrated relatively fast and reversible response of the chain by measuring the transmissivity of the chain subjected to a wide spectrum of elastic vibrations. The experiments are carried out with high precision allowing detection of the amplitude of vibrations within a very small area of each scatterer. The obtained results can find applications in design of remote controlled acoustic filters. I consider the presented results as a high-quality research, which, however, does not fit the scope of Nature Communication. The authors do not propose a new mechanism of control of the band structure; they rather propose a useful modification of the already known methods.

Reply

We thank the reviewer for her/his positive evaluation on the quality of the work. We would like, however, to comment on her/his opinion about the novelty of our approach.

The reviewer underlines the thermo-responsivity features of the azo-loaded polymeric resin we employed for fabrication, while addressing the interaction with light exploited to control the acoustic transmission spectrum as a known “laser heating” effect. It should be first observed that the interaction of light with the azo-units and the corresponding mechanical effect on the surrounding (passive) crosslinked matrix is more complex than a simple thermal softening due to heating and remains to be completely understood. We recall that while the azobenzene cyclic photoisomerization certainly promotes an efficient energy transfer from radiation to matter, leading to a temperature increase, the steric hindrance difference between the trans- and cis- isomers is also effective in altering the overall mechanical properties (e.g. density and Young’s modulus) of the embedding matrix. For example, in a recent work (ref. [49] in the manuscript), an azo-loaded BEDA matrix has been demonstrated to exhibit a surprising photo-hardening effect when laser-irradiated at temperatures higher than the glass-transition temperature T_g . This demonstrates the occurrence of a pure photo-mechanical effect in which the energy release from the azos to the hosting matrix happens to be negligible. To explain this, we have explicitly commented this point on page 3 and have added two references [46,47] on the role of photoisomerization, at temperatures below the T_g , in producing a larger polymer softening as compared to thermal heating.

More importantly, the unique selectivity offered by the proposed light-induced modification of the Young’s modulus in a polymer that has very low thermal conductivity is a key novel mechanism that opens new possibilities to achieve tunability and possibly grading of the metamaterial properties.

Nevertheless, we acknowledge the relevance of the reviewer’s criticism and have therefore performed further extensive experimental and computational work in order to shed light on the thermal effects induced by the laser irradiation on the BEDA-MR structure. All additional results are now provided in an additional “Supplementary Information” document, and commented in the manuscript main text (pages 7 and 8). These results include:

- (i) thermal imaging of BEDA-MR samples under laser irradiation, which proves the local nature of the heating;
- (ii) measurements of BEDA-MR Young's modulus as a function of frequency and temperature with DMA, providing direct experimental evidence of the mechanical modification effects of the heating;
- (iii) temperature-controlled transmission spectra measurements, to illustrate the effect of global heating (as opposed to local irradiation) of the samples;
- (iv) 3D Finite Element simulations of elastic wave transmission through the LRMM with a cumulative softening of individual pillars, to confirm the proposed band-gap modification mechanism (see discussion below). Since these are linear elastic simulations, results also show the independence of the mechanism from heating-related viscoelastic effects.

We believe these new results have improved the scientific soundness of the paper and help the reader to gain a more comprehensive view of the mechanism underlying the photo-control of the LRMM acoustic properties, which has never been considered previously in amorphous printed polymers.

I do not agree with the authors' interpretation of the obtained results. The authors claim that heating of a single scatterer leads to extension of the band gap. This is not true. If elastic properties of a single element in a periodic chain are different from the others, this element is considered as a point defect in a periodic system. It is known that presence of a point defect leads to a discrete level within the band gap (so-called impurity level). If the level appears close to one of the band edges and the level width (due to dissipation) is sufficiently broad, it can be mistakenly interpreted as broadening of the band gap. There is one more reason for the observed extension of the gap. The band edges are not sharp because the chain contains only 8 periods. If the last element in the chain is removed it becomes shorter that leads to further degradation of the band gap. If the element is not removed completely but replaced by a defect, a Tamm state may appear in the spectrum. Since the chain is short the level associated with this state may be sufficiently broad that, again, can be interpreted as gap broadening. Additional study is necessary in order to understand what is the real physical nature of the observed lowering of acoustic transmissivity.

In conclusion, I do not recommend this paper for publication in Nature Communication since the presented results do not contain a new physical effect and the proposed explanation is confusing.

Reply

The first thing to be said is that the reviewer's comments concern the interpretation of the experimental observations, and not the observations themselves. We certainly acknowledge the relevance of the raised point and agree that local changes of the material elastic properties within a 1D periodic lattice may result in the formation of impurity levels (including Tamm states), similarly to defects in electromagnetic, photonic or solid state structure counterparts. However, Tamm states are produced by means of interface defects (e.g. at the lattice-lattice or lattice-outer medium interface) resulting in additional modes whose energy often falls within the bandgap. If the defect mode occurs close to the band edge, the overall band gap is actually reduced, as outlined in X. Mei et al, "Acoustic Tamm States in Double 1D Phononic Crystals", DOI: 10.1007/s11595-012-0468-5. In fact, such a defect mode would result in an increase of the transmitted power across the mode bandwidth. Instead, we observe a bandgap enlargement regardless of the position of the illuminated pillar (i.e. the position of the introduced defect), which suggests the limited relevance of defect mode mechanism in this case.

Our understanding of the mechanism, instead, involves the effect of local resonance of the individual softened elements. We should indeed recall that the presented LRMM is more complex than a simple periodic arrangement of broadband scatterers. As described in the manuscript, each pillar is actually an individual mechanical resonator with its own vibrational eigenmode spectrum, determined by its shape, size and material properties. The elastic wave transmission observed without illumination results from an interplay of both the individual spectral response and the overall spatial periodicity of the pillars. Thus, the emerging attenuation bands at different frequencies can be ascribed to either Bragg (periodicity) or non-Bragg (local resonances) effects (see e.g. Romero-Garcia et al., “Multi-resonant scatterers in sonic crystals: Locally multi-resonant acoustic metamaterial”, *Journal of Sound and Vibration* 332, 184-198 (2013)).

In the proposed LRMM, when a pillar is laser-illuminated, it becomes softer and its vibrational spectrum is altered accordingly (see Figure 2a). This has already been studied on the same material in ref. [49], where a BEDA-DR1M cantilever is shown to reversibly down-shift and widen its main resonant peak. Similarly, the overlap of the original lattice bandgap with the shifted resonance spectrum of the softened pillar(/s) (which dissipate energy at different frequencies) can give rise to a widened band gap of the entire structure.

To verify the assumed effect, we have additionally performed a set of 3D FE simulations (see Supplementary Material and main text pag.8), in which the elastic wave transmission is calculated through the LRMM while a progressive number of pillars undergo softening (the Young's modulus is reduced by 28%, as estimated experimentally).

The main outcome can be summarized as follows:

- (i) when softening is applied to all pillars (from #2 to #8), the bandgap is down-shifted in frequency;
- (ii) when only few pillars are softened (for example pillar #2 or pillars #2 and #3), a wider attenuation band is produced. The down-shift of the low-frequency edge and the up-shift of the high-frequency edge are ascribed to an additional energy absorption/scattering performed by the softened pillars over a new frequency range, according to their modified vibrational spectra;
- (iii) for elastic waves at frequencies outside the bandgap, the transmitted amplitude is not significantly attenuated, meaning that few softened pillars do not introduce a significant broadband damping making the overall transmission spectrum to sink.

These results point towards a mechanism mainly related to local resonance, although Bragg scattering effects cannot be ruled out. Further verification of this conclusion is under way on other configurations.

This additional discussion has been added in the manuscript in a specific paragraph (“In order to better investigate the mechanism exploited in the band gap modification [...] These results point towards a mechanism mainly related to local resonance, although Bragg scattering effects cannot be ruled out”), and the details of the additional measurements and simulations are given in the Supplementary Information.

Reviewer #2

Authors presents the first steps of stimuli responsive metamaterials. The paper is well written and consistent. From the physical point of view i do not see any mistake. I cannot judge on the chemistry but the obtained results confirm the efficiency of the procedure.

In overall this study is perfectly in time with the current state of art to reach time modulated active metamaterial. I do recommend the paper for publication.

Minor: they should discuss more the maximum modulation time of such structures and fix the typos in references.

Reply.

We thank the Reviewer for his/her appreciation of our work. We have fixed the typos in the References and added a sentence in the main text, in order to discuss the aspect of the maximum modulation time.

Reviewer #3

The authors describe a one-dimensional acoustic metastructure based on a 3D printed light-responsive patterned waveguide. Evidence is given on how to tune the width of the frequency band gap. Undoubtedly the paper is interesting and well written, but some figures are unclear and needs to be revised.

In Conclusion the manuscript can be considered for publication on Nature Communications but only after some mandatory major revisions according to the following critical points:

Figure 2a: the authors should define what is the “spectral weight” on the z axis of the figure.

Figure 2f: it is unclear. On the vertical axis two different quantities are mixed: Time [ms] and Array longitudinal axis [mm]. This may confuse the general reader.

Figure 3b: Dotted red line is interrupted. Moreover the difference among the colour curves are not reported in the caption. The authors should clarify this point.

Line 74: the authors should explain in the text the meaning of the quantity T_g

Reply.

We thank the Reviewer for his/her suggestions that helped us to improve the clarity of our manuscript and of the annexed figures. We have implemented all the revisions proposed.

REVIEWERS' COMMENTS:

Reviewer #1 (Remarks to the Author):

Additional study made by the authors regarding the nature of the bandgap broadening sounds convincing. While there are still some doubts regarding the reason of observed attenuation, namely, it could be due to increased dissipation of the heated pillars or really to bandgap widening, I agree that the proposed structure serves as a tunable acoustic device. It may be used for manipulation with elastic waves and possesses an obvious advantage with respect to operation time over other existing devices. I recommend the revised manuscript for publication in Nature Communications. There are a few minor points for the authors' attention:

1. Several times in the text the concept of dissipation (absorption) and attenuation due to bandgap are mentioned together. It is not clear what the authors mean in each case. It would be helpful if the authors separate these two different physical reasons or clearly mention that the main reason for attenuation is unknown.
2. In lines 258-260 a linear dependence on laser power is mentioned. I do not see where this linear dependence appears in Fig. 3c.
3. I may recommend adding arrows to the lines showing the sizes of the elements of the unit cell in Fig 1e.
4. In Fig. 1g the edge of the structure labeled "out" is really the input edge.

Arkadii Krokhin

Reviewer #3 (Remarks to the Author):

The manuscript after revisions has been much more improved, and can now be published in the present version

Reviewer #1

Additional study made by the authors regarding the nature of the bandgap broadening sounds convincing. While there are still some doubts regarding the reason of observed attenuation, namely, it could be due to increased dissipation of the heated pillars or really to bandgap widening, I agree that the proposed structure serves as a tunable acoustic device. It may be used for manipulation with elastic waves and possesses an obvious advantage with respect to operation time over other existing devices. I recommend the revised manuscript for publication in Nature Communications.

Reply

We thank the reviewer for the positive comments.

There are a few minor points for the authors' attention:

1. Several times in the text the concept of dissipation (absorption) and attenuation due to bandgap are mentioned together. It is not clear what the authors mean in each case. It would be helpful if the authors separate these two different physical reasons or clearly mention that the main reason for attenuation is unknown.

Reply

We take the opportunity of this letter to add some clarification about this issue.

In the manuscript, we use the term "absorption" (and related terms) in two contexts:

- (i) in the light-azo interaction (lines 88, 150, 282);*
- (ii) in the elastic wave-LRMM interaction (lines 115, 119).*

The Reviewer's observation is clearly referring to context (ii), as context (i) refers is related to optical absorption. In context (ii), we refer to an "absorption" effect meaning that the elastic energy associated to the propagating wave in the structure is partially transferred to the resonant pillars according to their eigenmode spectrum (lines 115 and 119). Therefore, we would like to indicate an increasing portion of energy continuously subtracted to the propagating guided mode. In this sense, the elastic wave reaches the end of the waveguide with an "attenuated" amplitude, depending on the frequency. Therefore, the term "attenuation" is used to indicate a decrease of the elastic wave amplitude transmitted through the structure. In addition to this, the periodic arrangement of the pillars itself also produces an "attenuation" of the transmitted wave.

As a consequence, the total "attenuation" of the propagating elastic wave is the global result of both the aforementioned effects.

To clarify this, in the text, we stated that "[...], these observations suggest that the underlying mechanism for the light-induced tuning of the band gap is strongly associated to a modification of the single pillar local resonances, which seems to prevail on other possible collective Bragg-type effects" (lines 190-193, pag.7)".

2. In lines 258-260 a linear dependence on laser power is mentioned. I do not see where this linear dependence appears in Fig. 3c.

Reply

In Figure 3c, the laser power has been changed from 50 mW to 200 mW in four steps. As the laser power changes in time, we measure the corresponding transmitted elastic wave amplitude $A(t)$ at the end of the waveguide. In the manuscript, we take the amplitude of the elastic wave (with no illumination) as a baseline $A_H=45$ mV. When a steady-state is reached at the different laser power, we take the corresponding elastic wave amplitude A_L and calculate the relative signal attenuation as $\Delta A/A_H=(A_H-A_L)/A_H$ (see table below).

Laser Power (mW)	50	100	150	200
A_L (mV)	40	35	26	20
$\Delta A/A_H$	$1.1 \cdot 10^{-1}$	$2.2 \cdot 10^{-1}$	$4.2 \cdot 10^{-1}$	$5.5 \cdot 10^{-1}$

Table 1: Laser power changes and the corresponding measured transmitted elastic wave amplitude $A(t)$ at the end of the waveguide.

Plotting $\Delta A/A_H$ as a function the laser power shows a rather linear dependence upon fitting. However, we must acknowledge that four experimental points are not enough to justify a robust fit. For this reason, we have smoothed our claim in the text by stating that “*the relative amplitude attenuation $\Delta A/A_H$ scales proportionally with the laser power*”, which is certainly demonstrated (lines 271-272, pag.10).

3. I may recommend adding arrows to the lines showing the sizes of the elements of the unit cell in Fig 1e.

Reply

Figure 1e has been modified according to the suggestion.

4. In Fig. 1g the edge of the structure labeled “out” is really the input edge.

Reply

The sequence of vibrational spectra presented in Figure 1g should be read from left to right, starting with the position of “pillar #1” and ending with the position “out”, i.e. slightly after “pillar #8”. Such a sequence is flipped with respect to the sketch shown in Figure 1d, where pillars from #1 to #8 are presented from the right to the left (note that the waveguide propagation axis “x” is oriented as the pillar numbering). To avoid any misunderstanding, each pillar is explicitly labelled in both figures.

Reviewer #3

The manuscript after revisions has been much more improved, and can now be published in the present version

Reply

We thank the reviewer for the positive outcome.